# Optimal dispatching of regional power grid considering vehicle network interaction

Yuanpeng Hua[1], Shiqian Wang[1], Yuanyuan Wang[1], Linru Zhang[2,3], Weiliang Liu[2,3]*

**1** Economic and Technical Research Institute of State Grid Henan Electric Power Company, Henan, China, **2** Department of Automation, North China Electric Power University, Hebei, China, **3** Baoding key Laboratory of State Detection and Optimization Regulation for Integrated Energy System, North China Electric Power University, Hebei, China

* zhanglinru22@163.com

**Data Availability Statement:** All relevant data are publicly available from the Dryad repository (https://doi.org/10.5061/dryad.np5hqc00s).

## Abstract

When large-scale electric vehicles are connected to the grid for unordered charging, it will seriously affect the stability and security of the power system. To solve this problem, this paper proposes a regional power network optimization scheduling method considering vehicle network interaction. Initially, based on the user behavior characteristics and charging and discharging characteristics of electric vehicles, a charging and discharging behavior model of electric vehicles was established. Based on the Monte Carlo sampling algorithm, the scheduling upper and lower limits of each scheduling cycle of electric vehicles were described, and the scheduling potential of each scheduling cycle of electric vehicles was obtained. Then, the electricity price is then used as an incentive parameter to guide EV users to charge during periods of low electricity prices and participate in discharge during periods of peak electricity prices. Aiming at the highest economic efficiency, the best consumption effect of new energy and the smoothest demand-side power curve of regional power grid, a three-objective optimal dispatching model was established. In the later stage, uncertainty factors are taken into consideration by introducing the concept of interval numbers, and an interval multi-objective optimization dispatching model is established. The two dispatching models are solved by NSGA-II algorithm and improved NSGA-II algorithm, and the Pareto solution set is obtained. Finally, based on the analytic Hierarchy Process (AHP), the optimal scheduling scheme is determined. The Monte Carlo sampling method is used to simulate the user side charging demand, and the effectiveness of this method is verified. In addition, the results of the interval multi-objective optimization model and the deterministic multi-objective optimization model are compared, and it is proved that the solution results of the interval multi-objective model are more adaptive, practical and robust to the uncertain factors.

## Introduction

The influx of a vast number of EVs accessing the grid may inevitably lead to a challenging situation of peak demand, putting a strain on grid stability. However, EVs can function as an

**Funding:** The authors received no specific funding for this work.

**Competing interests:** The authors have declared that no competing interests exist.

effective energy storage device, and through the optimization of vehicle networking scheduling strategy, vehicles can help the power grid to achieve peak regulation, frequency modulation and new energy consumption. Specifically, by guiding EVs to supply electricity to the grid during peak periods and charge during low peak periods, the stability and safety of grid operation can be improved [1].

Incentive based on electricity price is a common method for electric vehicles to participate in dispatching. Wei Wu [2], studied the economic value of electric vehicles connected to the grid, established a power supply cost model, and investigated the cost reduction under three charging operation modes: random charging, controlled charging, and vehicle-to-grid (V2G) charging. Literature [3, 4], based on the previous questionnaire results, the behavior reality of electric vehicle users and empirical deduction, an electric vehicle behavior probability model was established to describe the randomness and uncertainty behaviors of EV users in travel and behavior decision-making. A multi-objective optimal scheduling model is established with the economic benefits of EV users and the load side peak load and valley filling as objective functions. Pan Xiaotian [5] established a four-objective optimization scheduling model with the aim of stabilizing load fluctuations, minimizing user costs, and extending elastic travel time and charging state to the greatest extent. With this model, user needs can be better met by providing a sufficiently flexible state of charge with higher travel time. Literature [6–8] respectively considered the problem of electric buses participating in power grid dispatching and the problem of resource allocation of electric vehicle charging piles. Compared with ordinary private cars, electric buses have the characteristics of limited operating range, long charging time, complex grid characteristics, etc. By introducing multi-objective multi depot optimal scheduling model, the total operating cost and peak load generated by concurrent charging activities can be minimized. The uncertainty of EV user preferences and decisions may affect V2G scheduling, resulting in the imbalance between the electric vehicle's schedulable capacity and the required power. However, the charging pile resource allocation method proposed in this paper based on the two-stage classification and hierarchical scheduling framework can solve such problems in real time. Reference [9] investigated the participation of electric vehicles in the energy scheduling of virtual power plants. When the electric vehicle aggregator adopts the deterministic strategy and the virtual power plant adopts the stochastic strategy, the energy complementarity is realized and the overall operating economy is improved. Ju L [10] incorporated electric vehicles into carbon virtual power plants as a flexible resource and used the concept of electric vehicle aggregators to flexibly respond to grid operation requirements. Sheng [11],proposed a multi-time scale active distribution network (ADN) scheduling method, which includes backup coordination strategy and scheduling framework to improve the adaptive capacity of distribution network and reduce the impact of fluctuating power on the upstream transmission network. The backup coordination strategy can schedule available backup resources based on their temporal and spatial characteristics.

Literature [12, 13] puts forward reactive power optimization strategies for power grid including electric vehicles, and establishes a reactive power optimization model aiming at reducing voltage deviation and network loss, so as to reduce the operating pressure of traditional reactive power compensation equipment.

As the link between the electric grid and electric vehicle users, electric vehicle aggregators play a crucial role in coordinating the economic interests between the grid and users, and are indispensable participants in the interaction process of the vehicle network. In literature [14–16], aggregators focus on maximizing their own economic benefits, while also taking into account the demand response needs of users and the power grid, and they will participate in V2G as an auxiliary regulation method. In addition, the analysis of the scheduling potential of electric vehicles is the basis of the implementation of vehicle network interactive optimization

scheduling technology. Literature [4, 17, 18], has established a probability model for the spatio-temporal characteristics of electric vehicles considering various travel needs of users.

At present, there are limited studies considering uncertainty factors in V2G technology. Kong [19] proposed a bid-based double-layer multi-time scale scheduling method for multi-operator virtual power plants In the upper layer, a bidding equilibrium-based power allocation and internal pricing method for operators was proposed. The fluctuation cost coefficient was introduced to express the impact of the uncertainty of renewable energy generation on the bidding process.

The above research results are true and effective, and provided a good reference for the research in this paper. The above paper has conducted a detailed study on the potential of electric vehicle scheduling, fully considering the impact of electric vehicle participation in grid scheduling on the reliability and economy of grid operation, and has achieved good research results. However, the research in the above papers is generally one-sided and does not fully utilize the scheduling potential of electric vehicles, and the final scheduling results do not meet the actual user needs, making practical application difficult. We have developed an EV model in this study that can effectively and reasonably describe the charging and discharging characteristics and behavior of EVs, fully considering the goals and constraints that EVs need to achieve in participating in the power grid dispatch process, introducing uncertainty description, and establishing an interval multi-objective optimization model to make the obtained optimal dispatch scheme more reasonable and practical.

## Methods

### Model for temporal and spatial distribution characteristics of electric vehicle load

The model has been modified based on the probability distribution fitting of electric vehicle load time characteristics and spatial characteristics [4] using electric vehicle daily travel data from a certain district in Chongqing. The resulting model is as follows.

1. Parking time. Based on the statistical data, the approximation of parking duration is estimated to follow a log-normal distribution [4, 17], with a mean of $\mu_p = 2.8$ and a variance of $\sigma_p = 0.95$. The probability density function is given by the following equation:

$$f_p(x) = \frac{1}{x\sigma_p\sqrt{2\pi}}\exp\left[-\frac{(\ln x - \mu_p)^2}{2\sigma_p^2}\right] \tag{1}$$

2. Arrival/departure time. The start time of charging or discharging for electric vehicles at a charging station can be approximated as the entry time, which follows a normal distribution with a mean of $\mu_s = 17.5$ and a variance of $\sigma_s = 4.1$. The probability density function of electric vehicle arrival time is defined by the given equation.

$$f_s(x) = \begin{cases} \frac{1}{\sigma_s\sqrt{2\pi}}\exp\left[-\frac{(x-\mu_s)^2}{2\sigma_s^2}\right], & (\mu_s - 12) < x \leq 24 \\ \frac{1}{\sigma_s\sqrt{2\pi}}\exp\left[-\frac{(x+24-\mu_s)^2}{2\sigma_s^2}\right], & 0 < x \leq (\mu_s - 12) \end{cases} \tag{2}$$

Electric vehicle departure time from the station is given by the following equation:

$$t_q(x) = t_s(x) + t_p(x) \tag{3}$$

3. Daily driving mileage. The initial State of Charge (SOC) of an electric vehicle is determined by its daily driving mileage. By fitting data, it has been found that the daily driving mileage of electric vehicles follows a log-normal distribution with a mean value of $\mu_r = 3.4$ and a variance of $\sigma_r = 0.98$. The probability density function for this distribution is given as follows:

$$f_r(x) = \frac{1}{x\sigma_r\sqrt{2\pi}} \exp\left[-\frac{(\ln x - \mu_r)^2}{2\sigma_r^2}\right] \tag{4}$$

4. Planned charging power. In order to ensure that the EVs have the desired *SOC* by the expected departure time, it is necessary to calculate the planned charging power $P_{evplan}$ at time $t$, which can only participate in scheduling if it falls within the upper and lower limits of the EV charging power.

$$P^t_{evplan} = \frac{(SOC_q - SOC^t_{ev})C_e}{t_q - t_d} \tag{5}$$

In the equation, $SOC_q$ represents the expected *SOC* of the user when leaving the station, $SOC^t_{ev}$ represents the current *SOC*, $C_e$ represents the rated capacity, $t_q$ represents the user's expected departure time, and $t_d$ represents the current time.

5. Initial SOC.

$$SOC_{init} = SOC_q - \frac{d_r}{d_e} \times 100\% \tag{6}$$

In the equation, $SOC_{init}$ represents the initial *SOC* of the electric vehicle upon arrival at the charging station, $d_r$ represents the daily travel distance of the electric vehicle, and $d_e$ represents the rated travel distance of the electric vehicle.

## Analysis of dispatch potential for electric vehicles

Charging behavior of a single electric vehicle follows the parallelogram law [4, 17]. The charging or discharging power for each time period can be determined based on the actual scheduling needs between the upper and lower limits of scheduling, and the maximum waiting time is a measure of whether the electric vehicle can accept discharging scheduling at time $t$ (i.e., the electric vehicle must start charging when this time is reached).

In the Fig 1, $Cev$ represents the scheduling capacity.

$$t_{max} = t_q - \frac{C_{evq} - C_{evs}}{P_{charge}} \tag{7}$$

In the expression, $C_{evq}$ represents the desired electric quantity, and $C_{evs}$ represents the current electric quantity.

The scheduling potential of electric vehicles at time $t$ in a large-scale electric vehicle system refers to the total potential of dispatchable charging and discharging load for participating electric vehicles in the region. Based on the current total power value at time $t$, the upper and

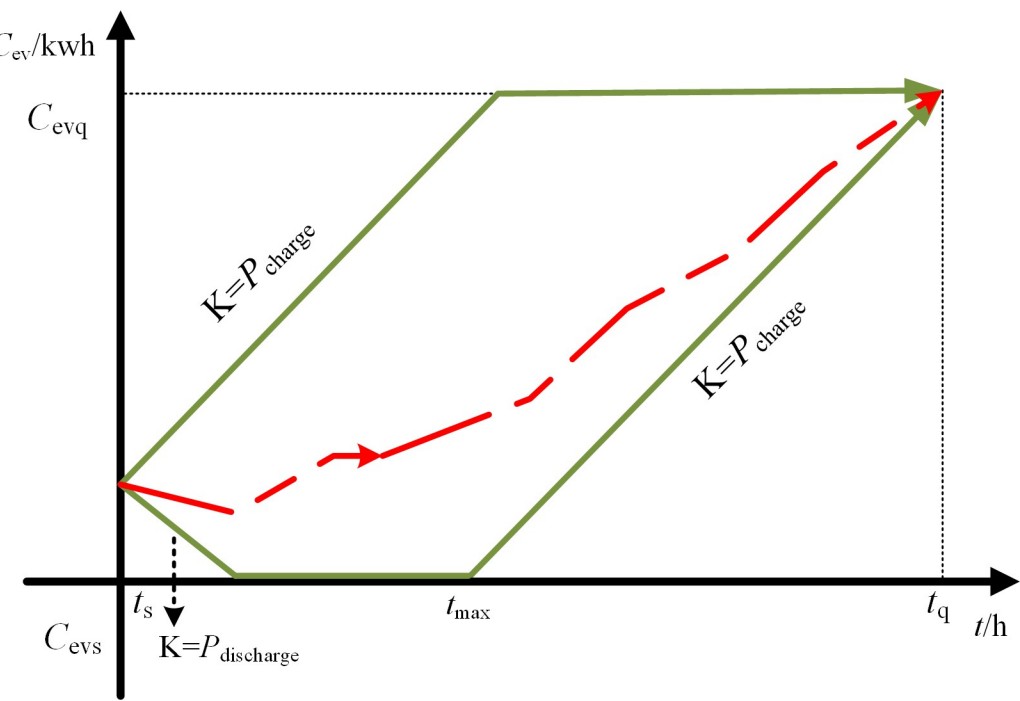

**Fig 1. Schematic diagram of EV scheduling potential.**

lower limits of dispatchable charging and discharging load at time $t+1$ can be predicted.

$$P_{ev\_max}^{t+1} = P_{ev}^t + \Delta P_{evcharge}^{t+1} + \Delta P_{evcharge\_c}^{t+1} \tag{8}$$

$$P_{ev\_min}^{t+1} = P_{ev}^t + \Delta P_{evcharge}^{t+1} + \Delta P_{evcharge\_d}^{t+1} \tag{9}$$

$$P_{ev\_min}^{t+1} \leq P_{ev}^t \leq P_{ev\_max}^{t+1} \tag{10}$$

In the expression, $P_{ev\_max}^{t+1}$ represents the upper limit of scheduling, $P_{ev\_min}^{t+1}$ represents the lower limit of scheduling, $P_{ev}^t$ represents the charging or discharging power of electric vehicles, $\Delta P_{evcharge}^{t+1}$ represents the additional EV load that must be added at time $t+1$ for scheduling, $\Delta P_{evcharge\_c}^{t+1}$ represents the EV load added only for charging scheduling, and $\Delta P_{evcharge\_d}^{t+1}$ represents the EV load reduced only for discharging scheduling.

### Multi-objective optimization scheduling model

**Parameters for optimization.** For a given scheduling period, let $x_s$ denote the decision variables for all scheduling time slots $\theta_T$ in the model. For time slot $t$, the decision variables $x_s$ are shown in Table 1.

**Objective function.**

1. Objective function for economic objective.
   The economic objective [19–21] function primarily considers the operational costs $C_{ag}$ of the regional power grid, including the cost of exchanging power with the external grid $C_{grid}$, the cost of wind power generation $C_w$, the cost of photovoltaic power generation $C_{pv}$, the cost of thermal power generation $C_f$, and the cost of energy storage system charging and

**Table 1. Decision variables.**

| Variable name | Interpretation |
|---|---|
| Pevcharge | Power exchange between electric vehicle and charging station |
| Pgrid | The regional power grid exchanges power with the external power grid |
| Pf | Thermal power |
| Pb | Charging and discharging power of electric energy storage |
| Pw | Real- time power of wind power generation |
| Ppv | Real- time power of photovoltaic power generation |

discharging $C_b$.

$$\min C_{ag}(x) = \min(C_{grid}(x) + C_w(x) + C_{pv}(x) + C_f(x) + C_b(x)) \tag{11}$$

$$C_{grid}(x) = \sum_{t=1}^{T} C_{ch\_grid}^t P_{grid}^t \tag{12}$$

$$C_w(x) = \sum_{t=1}^{T} C_{ch\_w}^t P_w^t \tag{13}$$

$$C_{pv}(x) = \sum_{t=1}^{T} C_{ch\_pv}^t P_{pv}^t \tag{14}$$

$$C_f(x) = \sum_{t=1}^{T} C_{ch\_f}^t P_f^t \tag{15}$$

$$C_b(x) = \sum_{t=1}^{T} C_{ch\_b}^t P_b^t \tag{16}$$

In Eqs 12–16, $C_{ch\_grid}^t$ represents the exchange power price between the regional power grid and external power grid, $C_{ch\_w}^t$ is the generation cost per unit of wind power generation, $C_{ch\_pv}^t$ is the generation cost per unit of photovoltaic power generation, $C_{ch\_f}^t$ is the generation cost per unit of thermal power generation, $C_{ch\_b}^t$ is the cost per unit of energy storage device for charging or discharging,.

2. Objective function for peak shaving and valley filling.

$$\min V_{load}(x) = \frac{1}{T} \sum_{t=1}^{T} (P_{load}^t - \bar{P}_{load})^2 \tag{17}$$

In the equation, $V_{load}$ represents the variance of the load curve, $P_{load}^t$ represents the power of in-network electricity consumption, and $\bar{P}_{load}$ represents the average load of electricity consumption.

3. Objective function for new energy integration.
   Due to the limitations of transmission lines and substations, there are constraints on the integration of new energy. The expression for the objective function for new energy

integration is as follows:

$$\Delta P_{\text{x}}^t = \begin{cases} (P_{\text{pv\_max}}^t - P_{\text{pv}}^t) + (P_{\text{w\_max}}^t - P_{\text{w}}^t), & P_{\text{pv\_max}}^t > P_{\text{pv}}^t \ \& \ P_{\text{w\_max}}^t > P_{\text{w}}^t \\ 0 & \text{else} \end{cases} \tag{18}$$

In the equation, $\Delta P_{\text{x}}$ represents the curtailed wind and photovoltaic power generation at a certain time, $P_{\text{pv\_max}}^t$ represents the maximum power generation of the photovoltaic units at time $t$, $P_{\text{w\_max}}^t$ represents the maximum power generation of the wind power units at time $t$.

**Constraints.** For $\forall t \in \theta_{\text{T}}$, the variables should satisfy the following constraints.

1) Constraints on power balance [21].

$$P_{\text{grid}}^t = P_{\text{Imload}}^t + P_{\text{ev}}^t - P_{\text{w}}^t - P_{\text{pv}}^t - P_{\text{f}}^t + P_{\text{b}}^t \tag{19}$$

In the equation, $P_{\text{Imload}}^t$ represents the conventional electricity load at time $t$.

2) Constraints on electric vehicle dispatch potential.

$$P_{\text{ev\_low}}^t \leq P_{\text{ev}}^t \leq P_{\text{ev\_high}}^t \tag{20}$$

In the equation, $P_{\text{ev\_low}}^t$ represents the lower limit power of electric vehicle charging or discharging scheduling during time period $t$, and $P_{\text{ev\_high}}^t$ represents the upper limit power of electric vehicle charging or discharging scheduling during time period $t$.

3) Constraints on upper and lower power limit for each device [21].

$$\begin{cases} P_{\text{grid\_min}} \leq P_{\text{grid}}^t \leq P_{\text{grid\_max}} \\ P_{\text{ev\_min}} \leq P_{\text{ev}}^t \leq P_{\text{ev\_max}} \\ P_{\text{b\_min}} \leq P_{\text{b}}^t \leq P_{\text{b\_max}} \\ P_{\text{f\_min}} \leq P_{\text{f}}^t \leq P_{\text{f\_max}} \\ 0 \leq P_{\text{w}}^t \leq P_{\text{w\_max}} \\ 0 \leq P_{\text{pv}}^t \leq P_{\text{pv\_max}} \end{cases} \tag{21}$$

In the equation, $P_{\text{grid\_min}}$ represents the lower limit of power exchanged with the external power grid, $P_{\text{grid\_max}}$ represents the upper limit of power exchanged with the external power grid, $P_{\text{ev\_min}}$ represents the lower limit of electric vehicle charging and discharging power, $P_{\text{ev\_max}}$ represents the upper limit of electric vehicle charging and discharging power, $P_{\text{b\_min}}$ represents the lower limit of battery charging and discharging power, $P_{\text{b\_max}}$ represents the upper limit of battery charging and discharging power, $P_{\text{f\_min}}$ represents the lower limit of thermal power generation, $P_{\text{f\_max}}$ represents the upper limit of thermal power generation, $P_{\text{w\_max}}$ represents the upper limit of wind power generation, and $P_{\text{pv\_max}}$ represents the upper limit of photovoltaic power generation.

4) Ramp rate constraint [22] for thermal power generation units.

$$P_{\text{down\_max}} \leq P_{\text{f}}^{t+1} - P_{\text{f}}^t \leq P_{\text{up\_max}} \tag{22}$$

In the equation, $P_{\text{down\_max}}$ represents the maximum ramp rate constraint for thermal power generation units in the downward direction, and $P_{\text{up\_max}}$ represents the maximum ramp rate constraint for thermal power generation units in the upward direction.

5) State of charge constraint [20, 21] for energy storage devices

$$\begin{cases} SOC_{\mathrm{T}} = SOC_{\mathrm{init}} \\ SOC_{\mathrm{b\_min}} \leq SOC_{\mathrm{b}}^{t} \leq SOC_{\mathrm{b\_max}} \end{cases} \tag{23}$$

In the equation, $SOC_{\mathrm{T}}$ represents the final value of $SOC$ for energy storage devices, $SOC_{\mathrm{init}}$ represents the initial value of $SOC$ for energy storage devices, $SOC_{\mathrm{b\_min}}$ represents the lower limit of $SOC$ for energy storage devices, and $SOC_{\mathrm{b\_max}}$ represents the upper limit of $SOC$ for energy storage devices.

## Interval multi-objective optimization scheduling model

**Interval optimization theory.**   The multi-objective optimization [23, 24] problem with intervals can be explicitly expressed as follows.

$$\begin{cases} \min_{x \in \Omega} F(x) = (f_1^I(x,c), f_2^I(x,c), \cdots f_M^I(x,c)) \\ s.t. \quad g_j(x,c) \geq q_j = [a_j, b_j], \quad j = 1, 2, 3, \cdots, m \\ \qquad h_k(x,c) = r_k = [a_k, b_k], \quad k = 1, 2, 3, \cdots, n \\ \qquad x \in \Omega \end{cases} \tag{24}$$

In the equation, $x$ is the decision variable, $\Omega$ is the decision space, $c$ is the interval vector, $g_j(x,c) \geq q_j$ represents the interval inequality constraint, and $h_k(x,c) = r_k$ represents the interval equality constraint, $f_i^I(x,c)$ is one of the objective functions, as it contains an interval vector $c$, and its value is also an interval number.

**Description of uncertainty factors.**   The uncertainty and random fluctuation range of wind power generation [25, 26], photovoltaic power generation [26, 27], and electrical load power are described by the interval numbers $[P_{\mathrm{w}}]$, $[P_{\mathrm{pv}}]$, and $[P_{\mathrm{Imload}}]$, respectively. Through probabilistic analysis, the aforementioned uncertain factors randomly fluctuate around their predicted values, and these fluctuations are symmetrical.

$$[P_{\mathrm{w}}^t] = [P_{\mathrm{w\_min}}^t, P_{\mathrm{w\_max}}^t] \tag{25}$$

$$[P_{\mathrm{pv}}^t] = [P_{\mathrm{pv\_min}}^t, P_{\mathrm{pv\_max}}^t] \tag{26}$$

$$[P_{\mathrm{load}}^t] = [P_{\mathrm{load\_min}}^t, P_{\mathrm{load\_max}}^t] \tag{27}$$

**Outer optimization model.**   The outer optimization model [28, 29] is usually used to determine the range of decision variables, with the goal of finding the optimal range of decision variables that enables the inner optimization model to optimize within this range and obtain the optimal solution. The objective function can be represented as follows.

$$\min_{x \in \Omega} F_s(x) = (f_{\mathrm{ag}}^I(x_s, c), f_{\mathrm{load}}^I(x_s, c), f_{\mathrm{x}}^I(x_s, c)) \tag{28}$$

The constraints of the outer optimization model are basically the same as those in the 'Constraints' section, and the constraints involving the number of intervals are as follows, for $\forall t \in \theta_{\mathrm{T}}$:

1) Power balance.

$$[P_{\text{grid}}^t] = [P_{\text{Imload}}^t] + P_{\text{ev}}^t - [P_{\text{w}}^t] - [P_{\text{pv}}^t] - P_{\text{f}}^t + P_{\text{b}}^t \tag{29}$$

2) Power constraints for each device.

$$\begin{cases} [P_{\text{w}}(t)] \in (P_{\text{w\_min}}, P_{\text{w\_max}}) \\ [P_{\text{pv}}(t)] \in (P_{\text{pv\_min}}, P_{\text{pv\_max}}) \end{cases} \tag{30}$$

**Inner optimization model.**   The inner optimization model [28, 29] optimizes within the range of decision variables $x^*$ s determined by the given outer optimization model to achieve optimization objectives.

$$f_{\text{ag}}^I(x_s^*, c) = [\min_{y \in c} f_{\text{ag}}(x_s^*, y), \max_{y \in c} f_{\text{ag}}(x_s^*, y)] \tag{31}$$

$$f_{\text{load}}^I(x_s^*, c) = [\min_{y \in c} f_{\text{load}}(x_s^*, y), \max_{y \in c} f_{\text{load}}(x_s^*, y)] \tag{32}$$

$$f_{\text{x}}^I(x_s^*, c) = [\min_{y \in c} f_{\text{x}}(x_s^*, y), \max_{y \in c} f_{\text{x}}(x_s^*, y)] \tag{33}$$

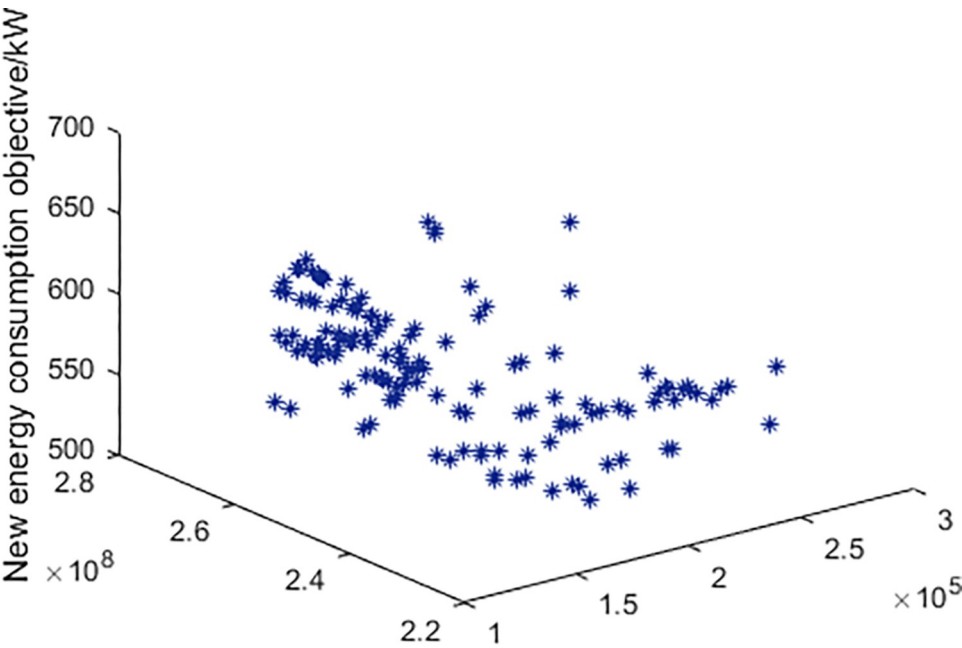

**Fig 2. The pareto frontier with the participation of electric vehicles.**

**Table 2. The objective function values of scheduling schemes considering extreme scenarios.**

| Objective functions | Economic objective | Peak shaving and valley filling objective | New energy consumption objective |
|---|---|---|---|
| Economic objective | 103173.15 | 2.69×108 | 533.33 |
| Peak shaving and valley filling objective | 215145.24 | 2.33×10$^8$ | 881.31 |
| New energy consumption objective | 215002.58 | 2.36×10$^8$ | 505.71 |

**Model solving.** This paper uses an improved NSGA-II algorithm to solve the problem of outer optimization model solving. By introducing interval credibility and interval overlap, it judges the situation where individuals meet the constraint conditions and compares individuals with the same rank to calculate individual crowding distance.

For the interval numbers [30, 31] $q_1 = [a_1, b_1]$, $q_2 = [a_2, b_2] \in I(R)$, the expression of interval credibility is as follows, where $w_1 = w(q_1)$ and $w_2 = w(q_2)$:

$$P(q_1 \geq q_2) \overset{\text{def}}{=} \max\left\{ 1 - \max\left\{ \frac{b_2 - a_1}{w_1 + w_2}, 0 \right\}, 0 \right\} \tag{34}$$

Eq (34) is recorded as the interval confidence level of $q_1 \geq q_2$.

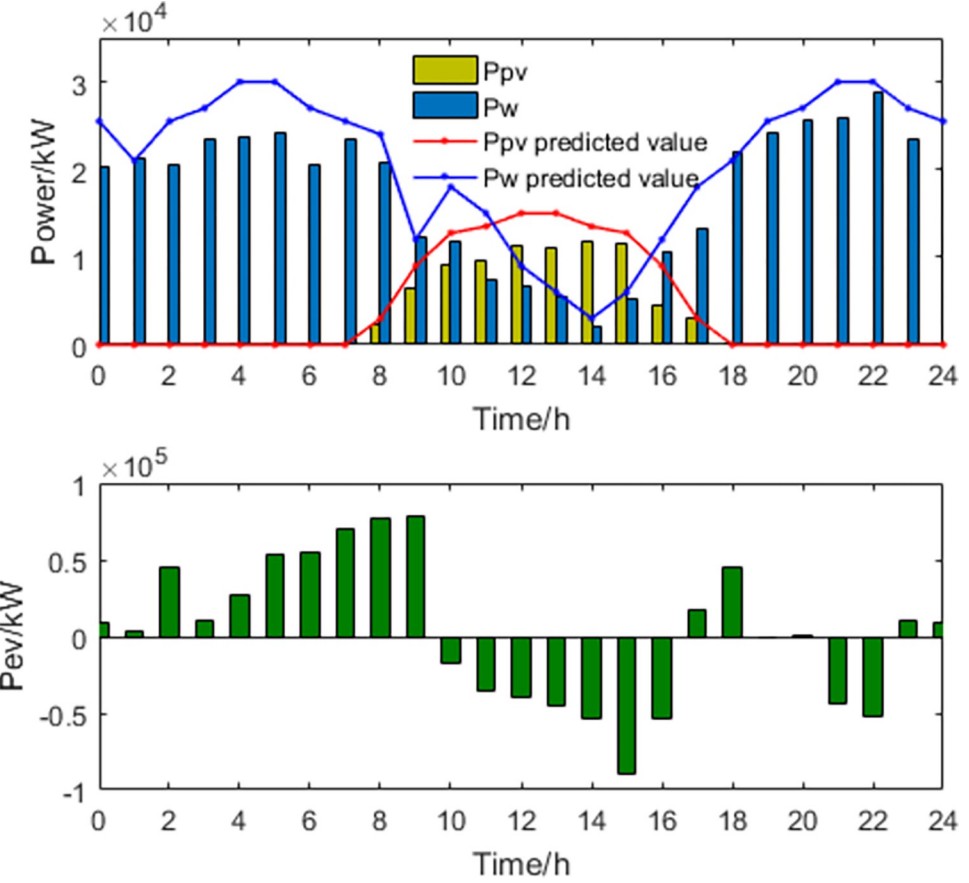

**Fig 3. New energy generation power and EV power.**

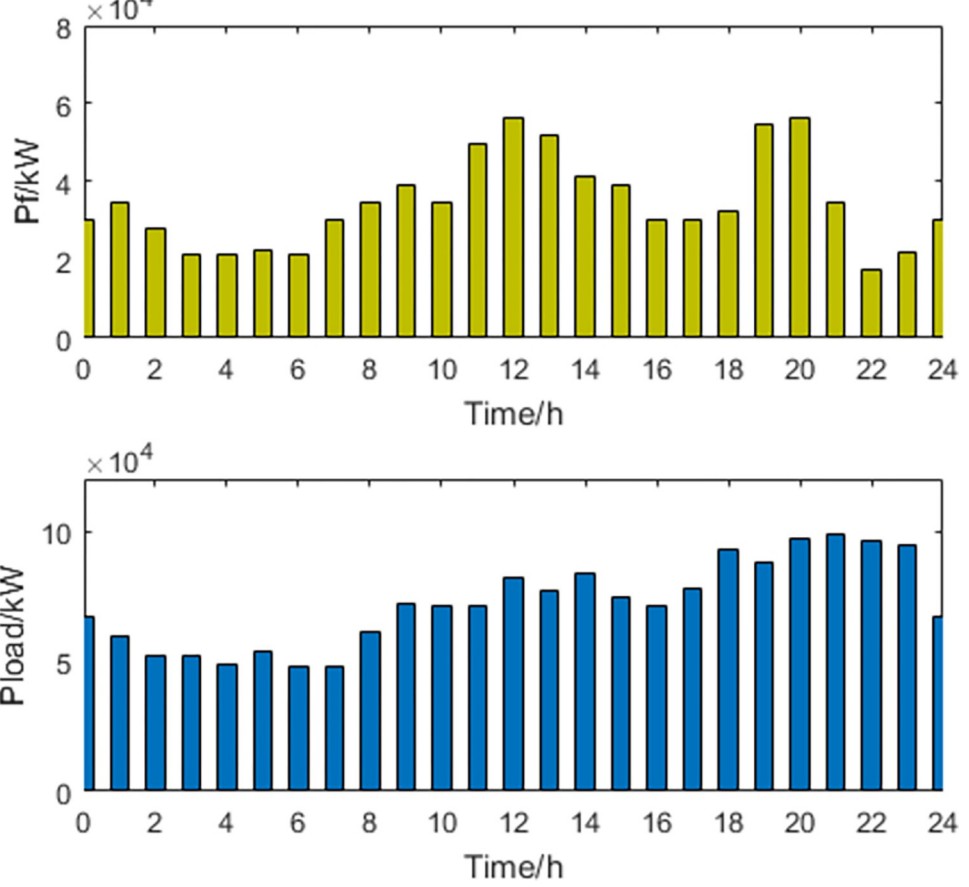

**Fig 4. Thermal power generation and load power curve.**

For an individual $x$, if its interval confidence level satisfying the j-th constraint is $\xi_j$, then:

$$\xi_j = P(g_j(x, c) \geq a_j) \tag{35}$$

Correspondingly, the degree of violation $L_j$ of the j-th constraint for individual $x$ is:

$$L_j = 1 - P(g_j(x, c) \geq a_j) = P(g_j(x, c) \leq a_j) \tag{36}$$

By setting the confidence threshold $\xi^*\,j$, an individual $x$ is considered feasible if its interval confidence level for a certain constraint is greater than or equal to this threshold, otherwise, it is considered infeasible. The dominance relation is determined by comparing the dominance relationships between feasible solutions and infeasible solutions. For evolutionary individuals $x_1$ and $x_2$ with the same rank value, the intersection of their objective function values can be represented by $f_i(x_1,c) \cap f_i(x_2,c)$. The interval overlap degree between $x_1$ and $x_2$ is expressed as:

$$\varphi(x_1, x_2) = \prod_{i=1}^{M} w(f_i(x_1, c) \cap f_i(x_2, c)) \tag{37}$$

## Results and discussion

The regional power grid in a certain urban area of a city was selected as the research object, and a case study was conducted on the energy optimization and scheduling of the regional

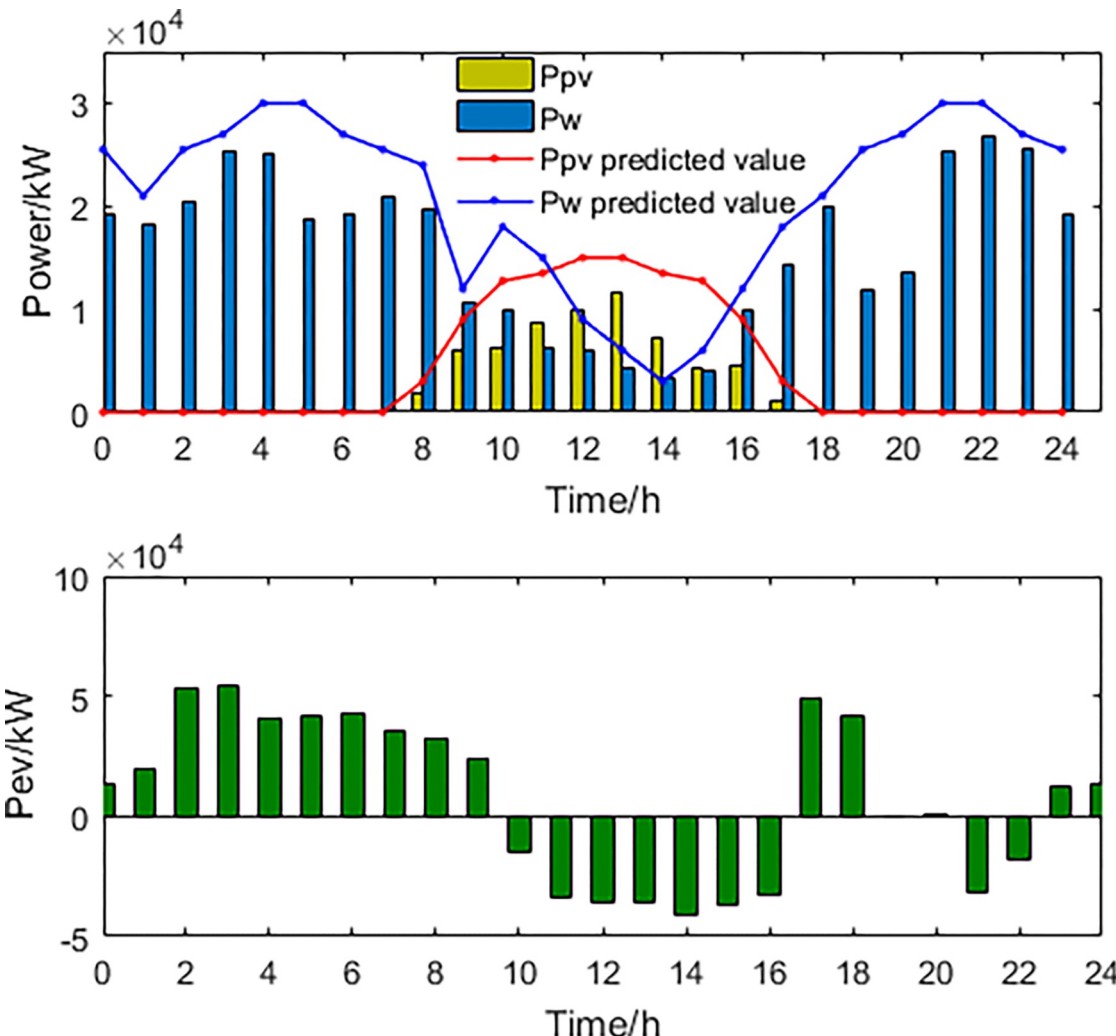

**Fig 5. New energy generation power and EV power.**

power grid. The 224-hour day was divided into 24 scheduling periods, with each hour serving as a scheduling period. In the case study, the wind power unit in the regional power grid was configured as 30 MW, the photovoltaic unit was 15 MW, the thermal power unit was 65 MW, the energy storage equipment was 10 MWh, and approximately 10,000 electric vehicles participated in the scheduling. The exchange power limit of the regional power grid's purchase and sale of electricity to the external power grid is ±60 MW.

The NSGA-II algorithm is set with the population size of $N_{pop} = 200$, the maximum number of iterations of $g_{max} = 10000$, the crossover probability of $P_c = 0.9$, and the mutation probability of $P_m = 0.2$.

## EVs participating in the scheduling

In the Pareto frontier, the distribution of objective function values is shown in Fig 2, where each point represents a Pareto optimal solution.

Three optimization and scheduling schemes were selected for analysis based on their economic viability, high renewable energy absorption capacity, and effective peak-shaving and

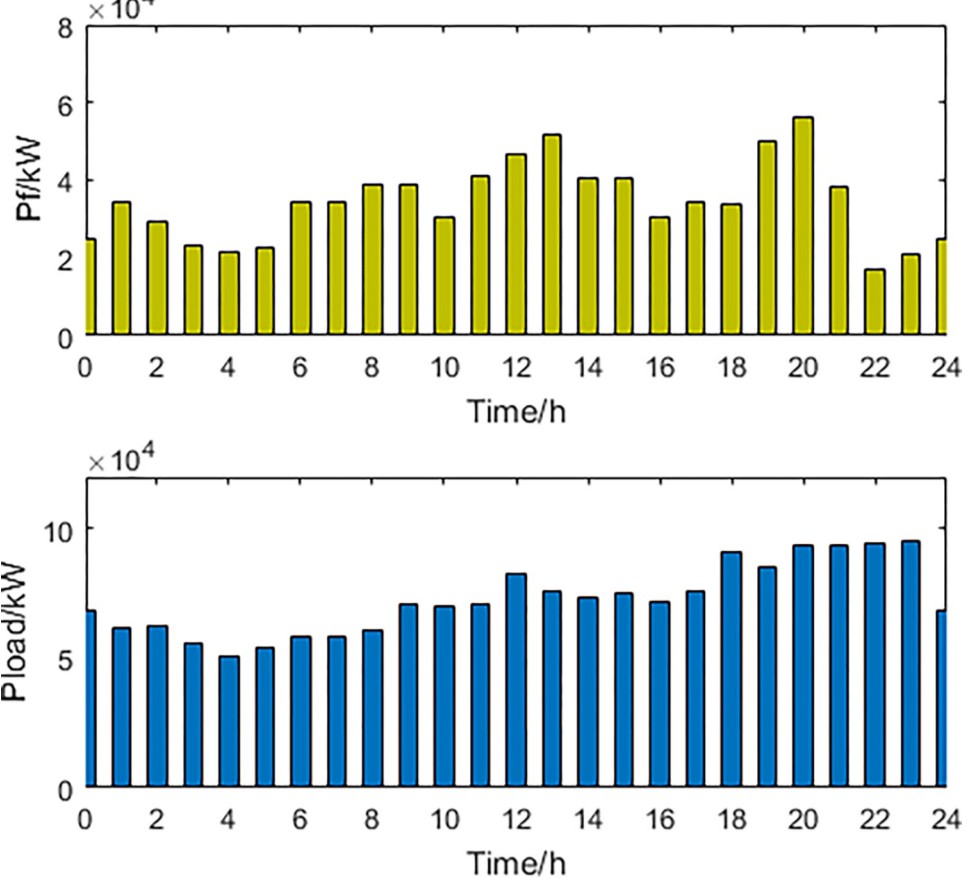

**Fig 6. Thermal power generation and load power curve.**

valley-filling effects. The most user-demand-oriented optimization and scheduling scheme was determined using the AHP decision-making analysis method. The objective function values under different scheduling schemes are shown in Table 2. Clearly, due to the lower cost of renewable energy generation compared to thermal power generation, the economic index of the power grid and the peak-shaving and valley-filling indexes and renewable energy absorption capacity index exhibit a trend of trade-off.

Plot the three optimization and scheduling schemes that are the most economically viable, have the best peak-shaving and valley-filling effects, and have the highest renewable energy absorption capacity.

**1) The best economic dispatch results**. The power curves of the most economical scheduling scheme are shown in Figs 3 and 4.

In the most economically efficient scheduling plan, during periods of high electricity prices and peak usage, the regional power grid needs to sell electricity to the large grid to meet the demand, resulting in an increase in power generation from wind, solar, and thermal sources. However, due to the limitations of power transmission capacity, some of the power generated from wind and solar sources may be wasted, resulting in a waste of energy resources. Conversely, during periods of low electricity prices and low usage, the regional power grid needs to buy more electricity from the large grid, resulting in a decrease in power generation from wind, solar, and thermal sources. As energy storage devices have lower participation costs in

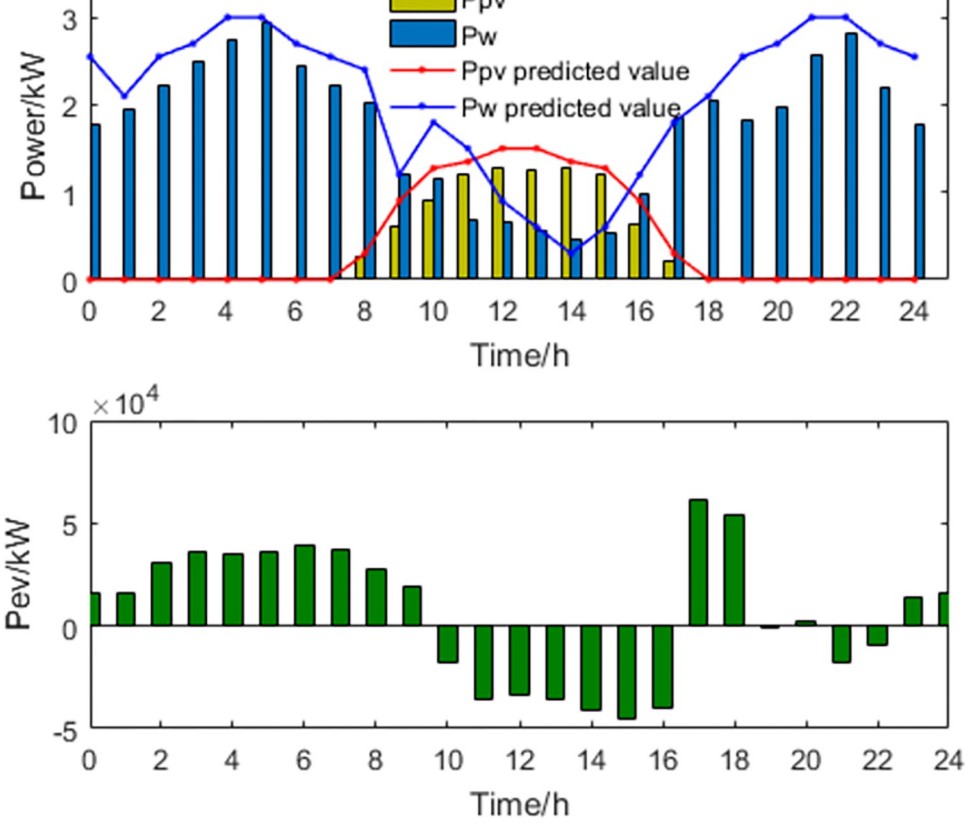

**Fig 7. New energy generation power and EV power.**

scheduling, they are used more during this time period, while the participation power of EVs decreases. This may cause a decrease in the system's ability to balance peak and off-peak demand, as the scheduling capacity of energy storage devices is limited and may not be able to meet the demands during peak periods.

**2) The results of the optimal peak-shaving and valley-filling scheme**. The power curves of the optimal scheduling scheme for peak shaving and valley filling are shown in Figs 5 and 6.

For the optimal peak-shaving and valley-filling scheme, in order to make the load curve as smooth as possible and reduce the peak-to-valley difference, EVs and energy storage devices are charged at high power during periods of low electricity demand and discharged at high power during periods of high electricity demand. As both thermal power generation and energy storage systems can control their power output, they can adjust their generation capacity flexibly to provide additional electricity during peak load periods and reduce the load during off-peak periods. However, this will increase the involvement of thermal power generation in load regulation, which will lead to a decrease in economic goals and a lower capacity for integrating new energy sources.

**3) The results of the scheme with the highest capacity for accommodating new energy consumption**. The power curves of the scheduling scheme with the highest new energy consumption capacity are shown in Figs 7 and 8.

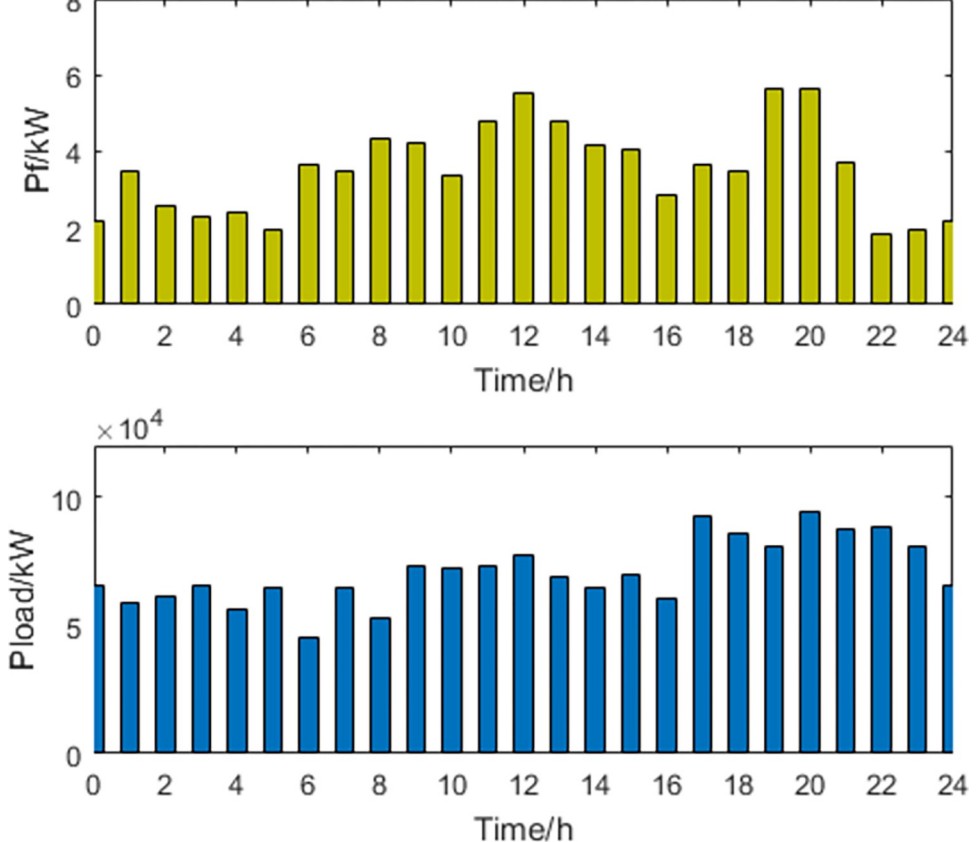

**Fig 8. Thermal power generation and load power curve.**

For the optimal strategy aimed at the highest new energy consumption capacity, the best approach is to maintain relatively high power generation of wind and photovoltaic power units at each scheduling time. However, since the output of wind and photovoltaic power generation fluctuates due to environmental factors, the peak-shaving and valley-filling effect of this approach is not ideal. In contrast, thermal power generation with higher output during the low valley of wind and photovoltaic power generation can compensate for the insufficient output of wind and photovoltaic power generation, while reducing output during the peak of wind power generation to avoid wasting resources. In addition, the power output of EVs participating in scheduling should have higher discharge power and lower charging power during the low valley of wind and photovoltaic power generation, and lower discharge power and higher charging power during the peak of wind and photovoltaic power generation.

Based on the importance of three decision factors, a pairwise comparison judgment matrix $A$ is constructed. After consistency check and hierarchical ranking, the normalized weight values of each index are $W = [0.1047\ 0.6370\ 0.2583]$. The optimal scheduling scheme is selected. The variable values of the optimal scheduling scheme are shown in Figs 9 and 10:

## EVs do not participate in scheduling

When electric vehicles do not participate in scheduling, we believe that they start charging when they enter the station, until the charging is completed or the electric vehicle leaves the

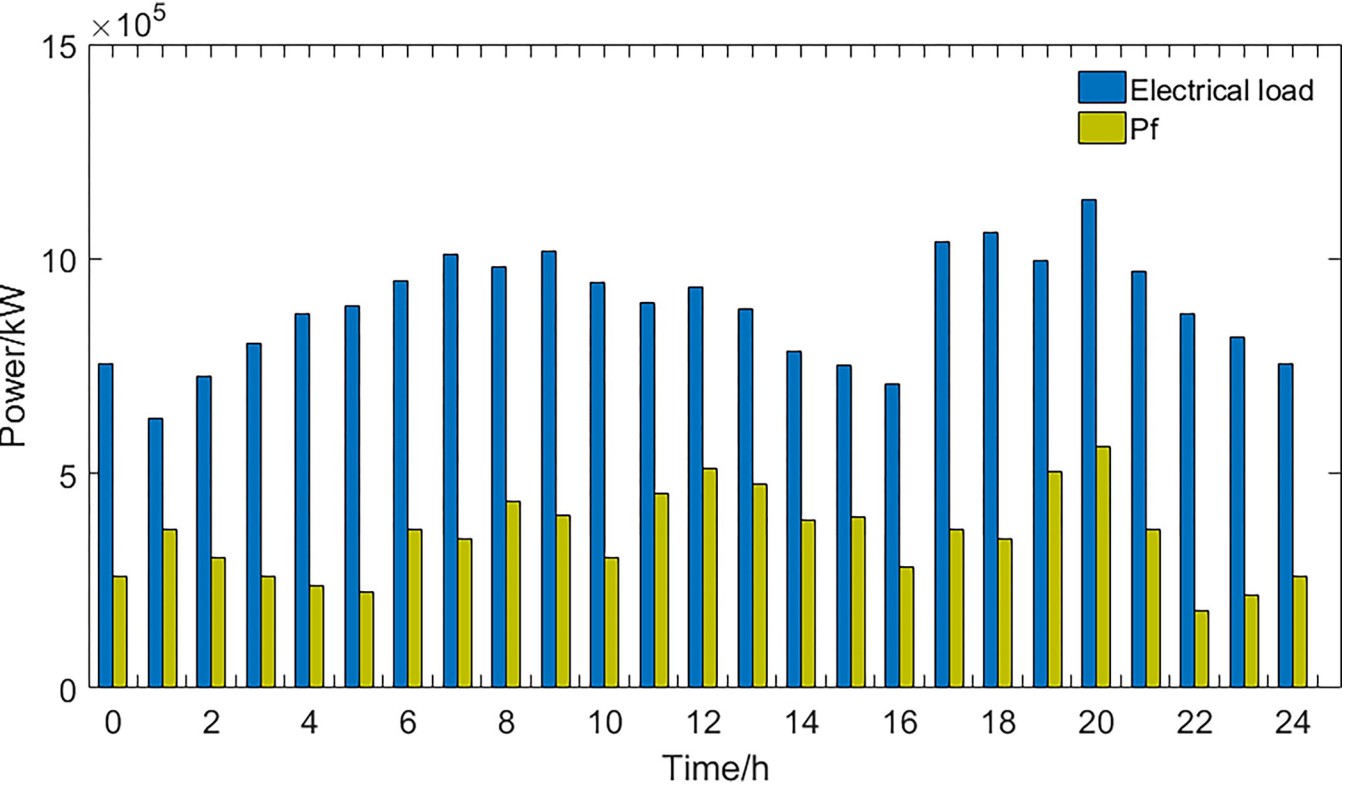

**Fig 9. Power output of thermal power units and load power.**

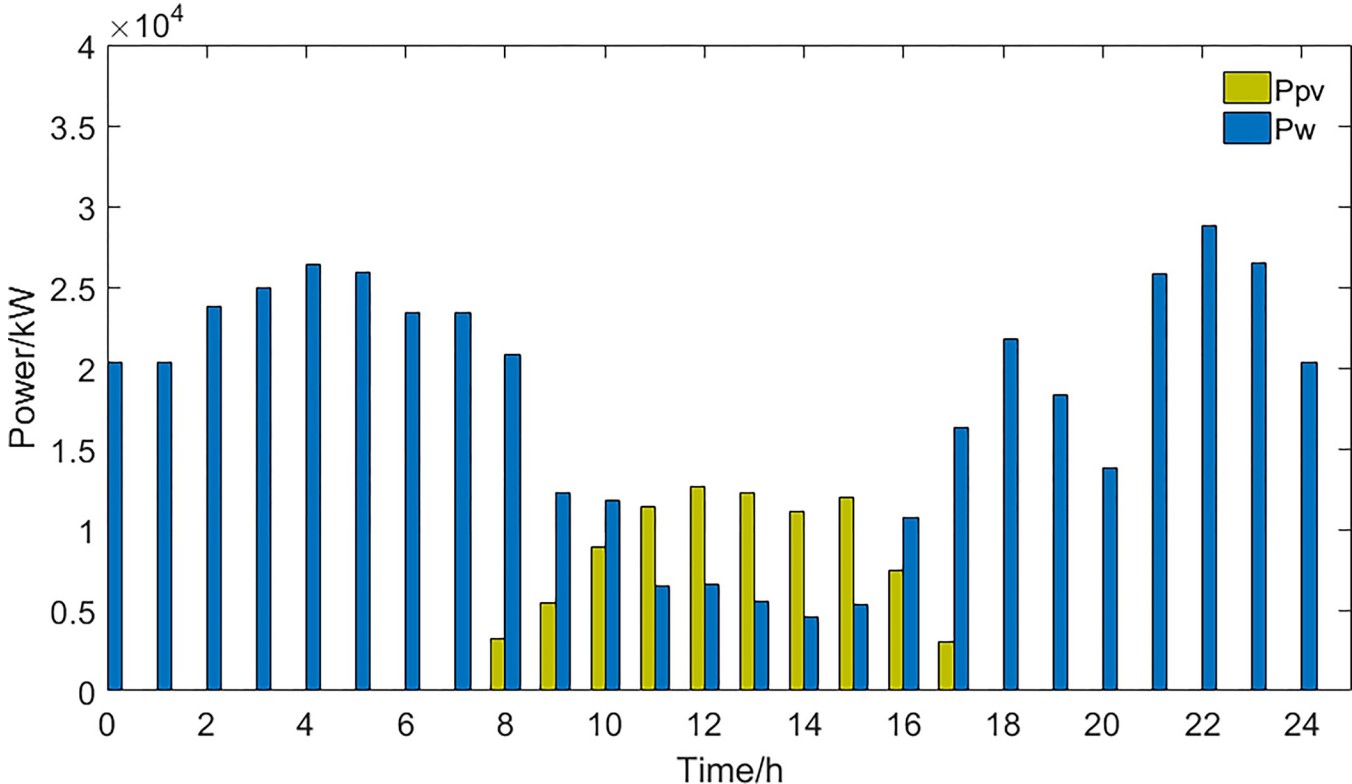

**Fig 10. Grid connected power of new energy generation.**

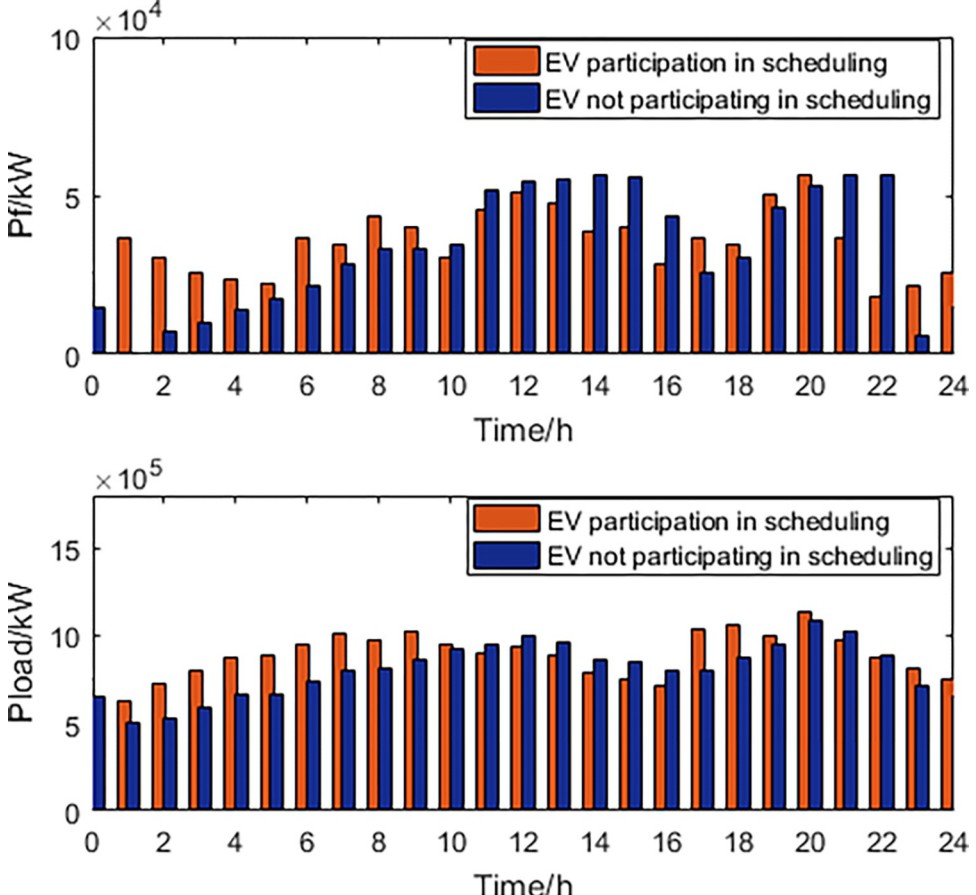

**Fig 11. Thermal power, load curve.**

station. Establishing a regional power grid scheduling model based on electric vehicles as regular loads that do not participate in scheduling, with the goal of power grid economics and new energy consumption as the objective functions. The decision variables include thermal power generation, photovoltaic power generation, and wind power generation, with the optimal Pareto solution set and frontier obtained. The optimal dispatching scheme is obtained based on the AHP. The comparison of dispatching results between the cases where EVs participate and do not participate in scheduling is shown in Figs 11 and 12.

By comparing the power generation of thermal power units, it is found that the participation of electric vehicles in scheduling reduces the power generation during peak electricity consumption, while increasing the power during off peak electricity consumption, ensuring the long-term stable operation of thermal power plants. Comparing the load curve, it is found that when electric vehicles participate in scheduling, the fluctuation rate of power load decreases by 38.21% compared to when electric vehicles do not participate in scheduling. Comparing the output power of wind turbines and photovoltaic power generation units, it is found that when electric vehicles participate in scheduling, the abandonment rate of wind and photovoltaic power generation decreases by 31.24% compared to when electric vehicles do not participate in scheduling. The optimized scheduling results obtained from the participation of EVs in the scheduling can be compared with the simulation results without the participation of EVs. It is evident that EVs have significant advantages in balancing electricity loads and

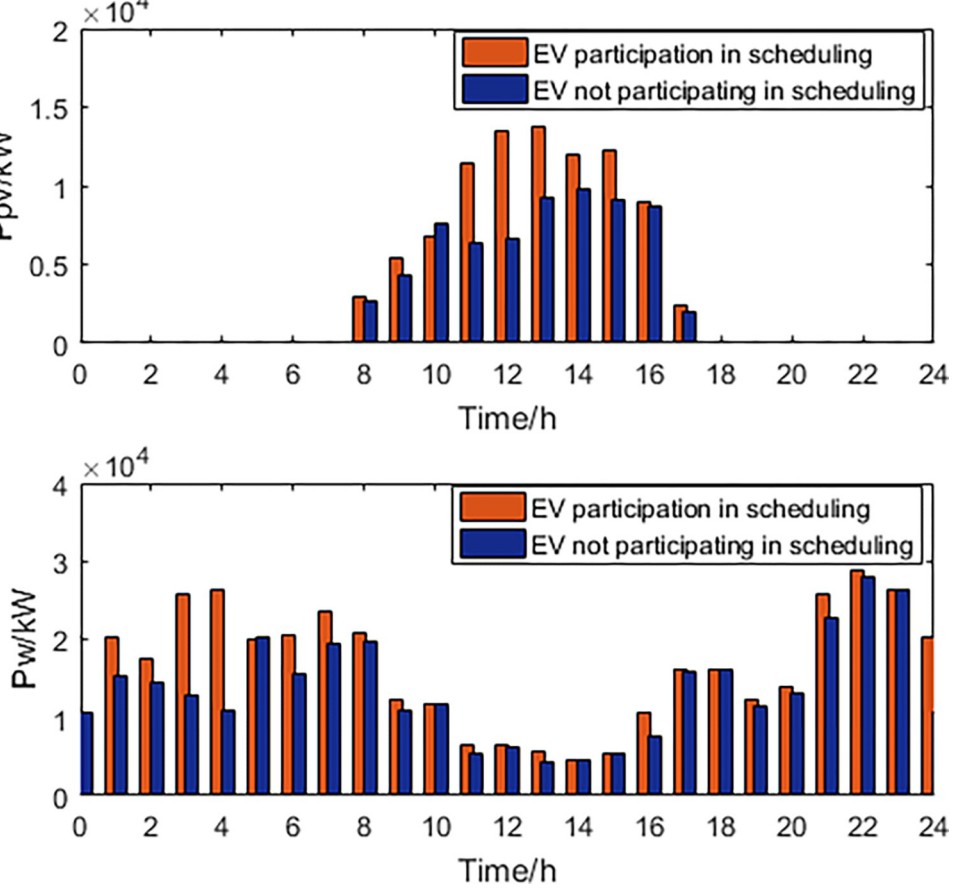

**Fig 12. New energy generation power.**

peak shaving. The fluctuation rate of electricity load is reduced by 38.21%. The participation of EVs in the scheduling reduces the output of thermal power generation during peak electricity consumption, while increasing its output during off-peak consumption, ensuring the long-term stable operation of the power plants. The abandonment rate of wind and PV power is reduced by 31.24%.

### Interval multi-objective optimization scheduling model solution

Configure the NSGA-II algorithm with the population size of $N_{pop} = 100$, the maximum number of iterations of $g_{max} = 10000$, the crossover probability of $P_c = 0.9$, and the mutation probability of $P_m = 0.1$.

Based on historical data analysis, the fluctuation range of predicted values for wind power, photovoltaic power generation, and conventional power load obtained through MATLAB simulation are shown in Figs 13–15.

Based on the above data and the improved NSGA-II algorithm, the interval multi-objective optimization model was solved to obtain the Pareto solution set in the objective functions, and the scatter plot of the median values in the objective function was drawn as shown in Fig 16. The distribution of each objective function value is shown in Figs 17–19.

Based on the AHP method for multi-objective decision-making, the optimal scheduling scheme was determined. The results are shown in Table 3.

Compare the scheduling results of deterministic optimization model and interval optimization model for the most economical scheduling scheme. Assuming the most economical

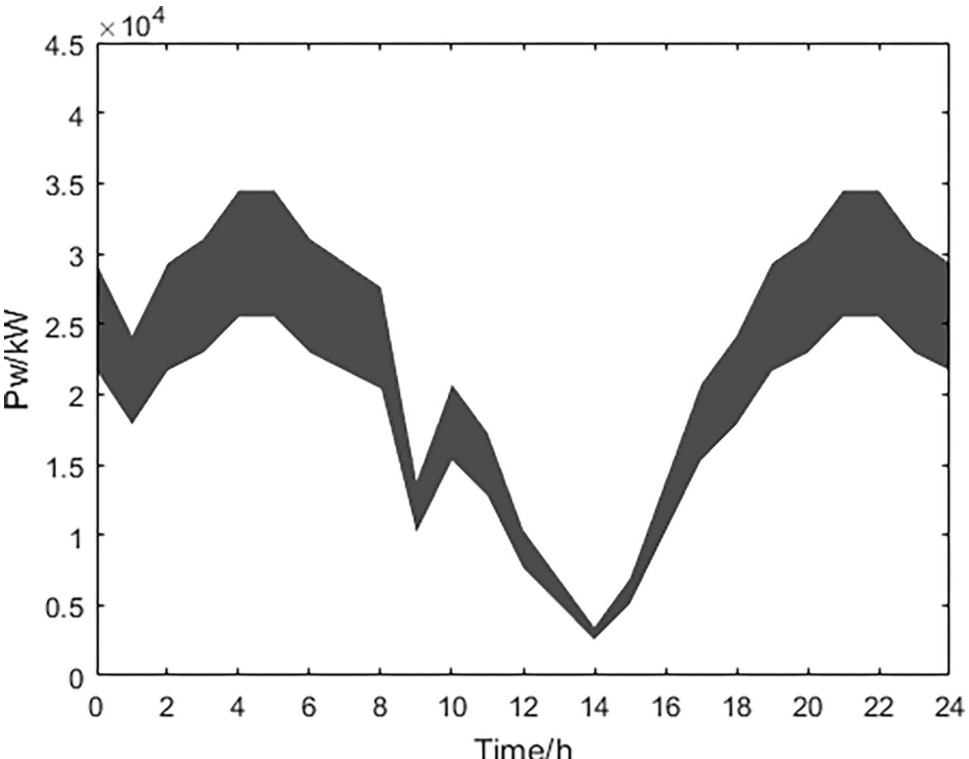

**Fig 13. Wind power prediction.**

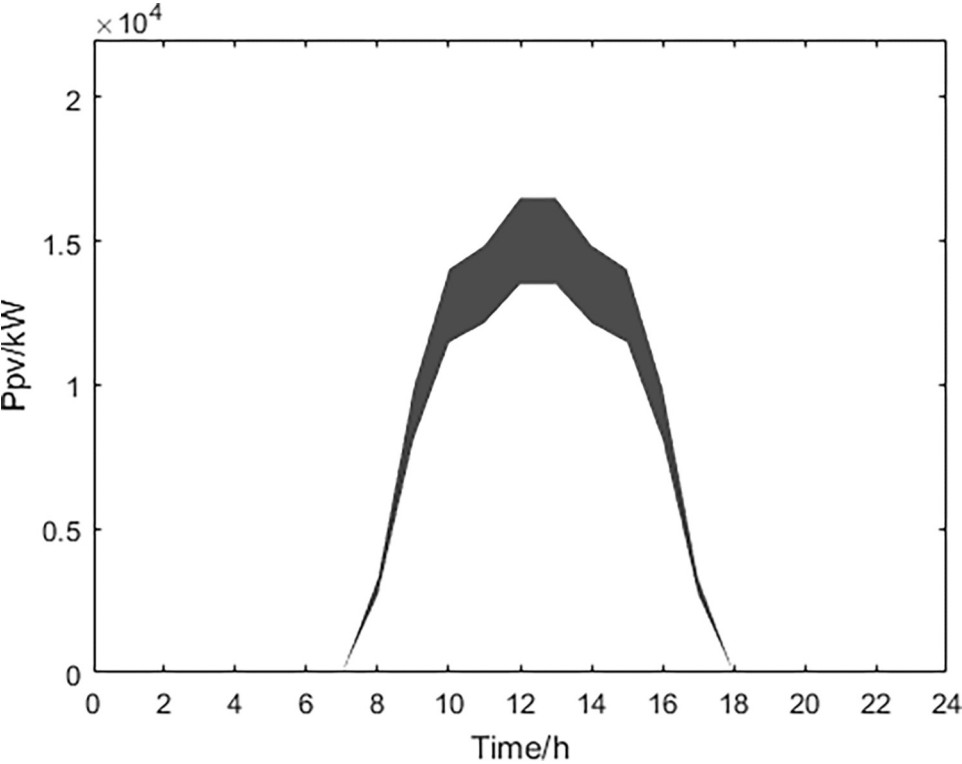

**Fig 14. Photovoltaic power prediction.**

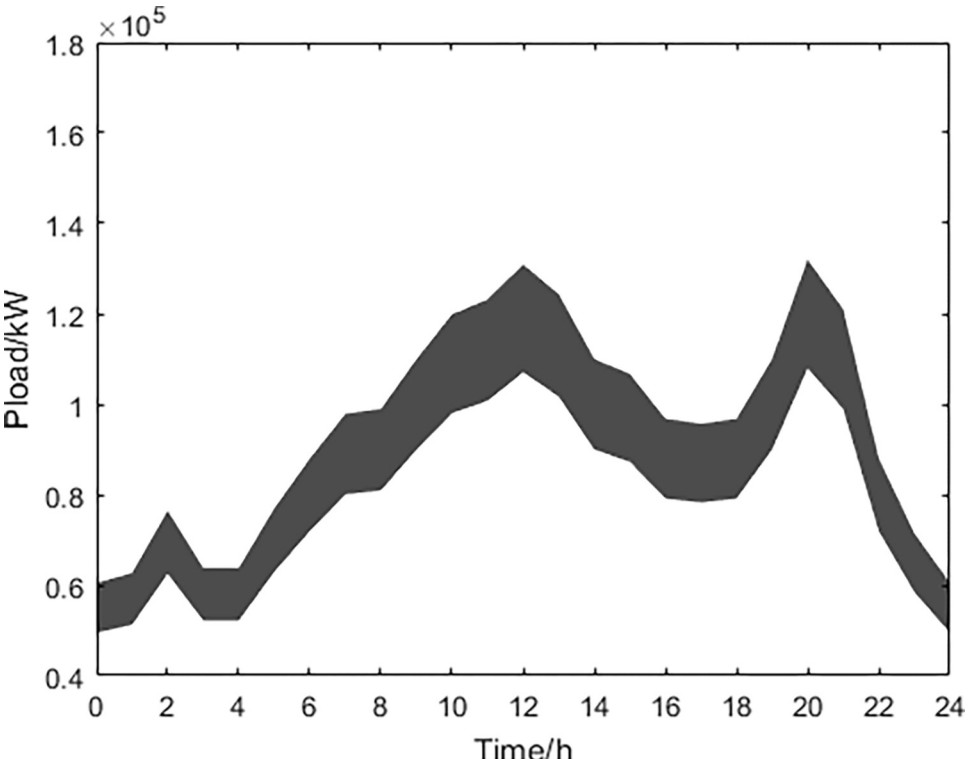

**Fig 15. Conventional electrical load power prediction.**

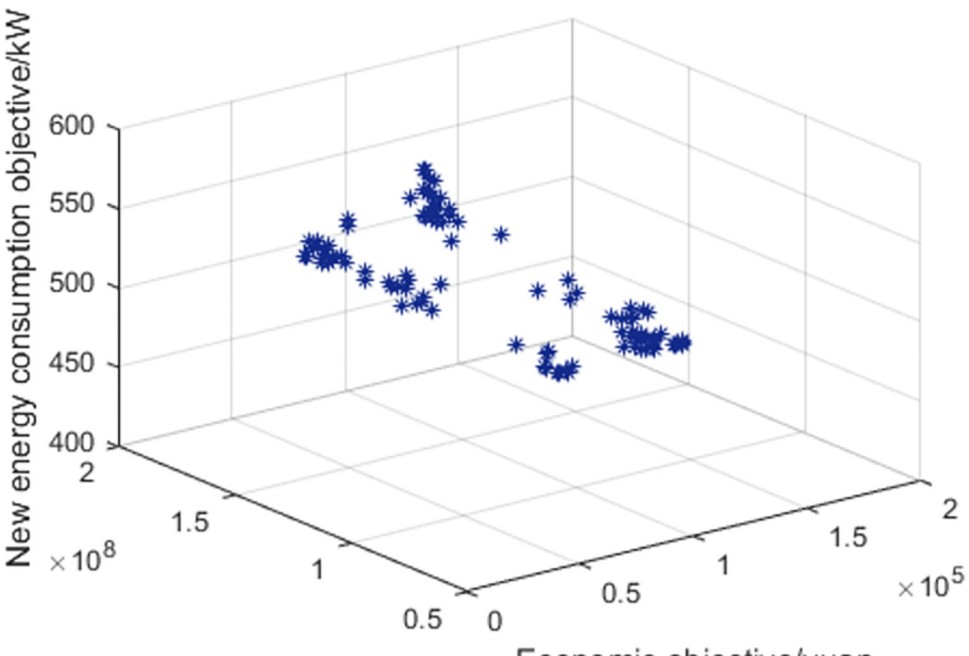

**Fig 16. Scatter plot of median values.** Incorporating uncertainty into interval multi-objective optimization models leads to a more reliable and robust set of solutions for the objective function values.

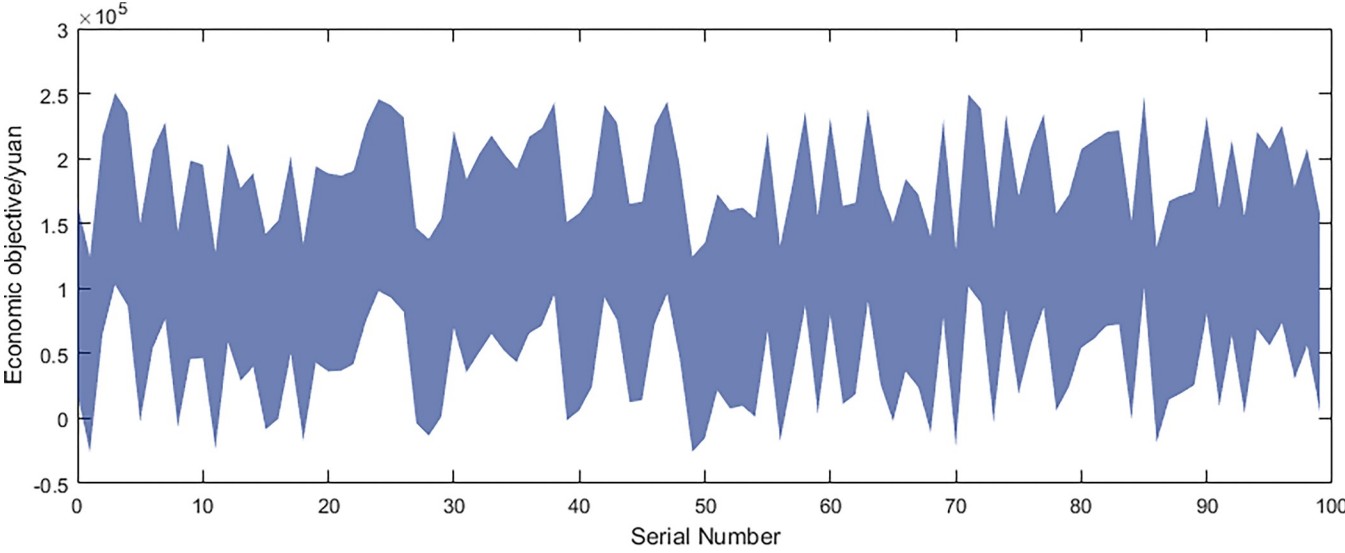

**Fig 17. Economic objective.**

scheme is applied to the actual power grid, considering the impact of uncertainty factors, mainly due to the smoothing of load power fluctuations by thermal power units. When the output of thermal power units is subject to ramping constraints and power upper and lower limits, the load power fluctuations will be smoothed by the transmission power between the external power grid. The objective function values under different fluctuation coefficients are shown in Tables 4 and 5.

Compared with the most economical solution obtained by the deterministic optimization model, the economic objective function interval of the most economical solution obtained by the interval optimization model has a larger median and a smaller interval width. The interval median and width of the peak shaving and valley filling objective function are both smaller.

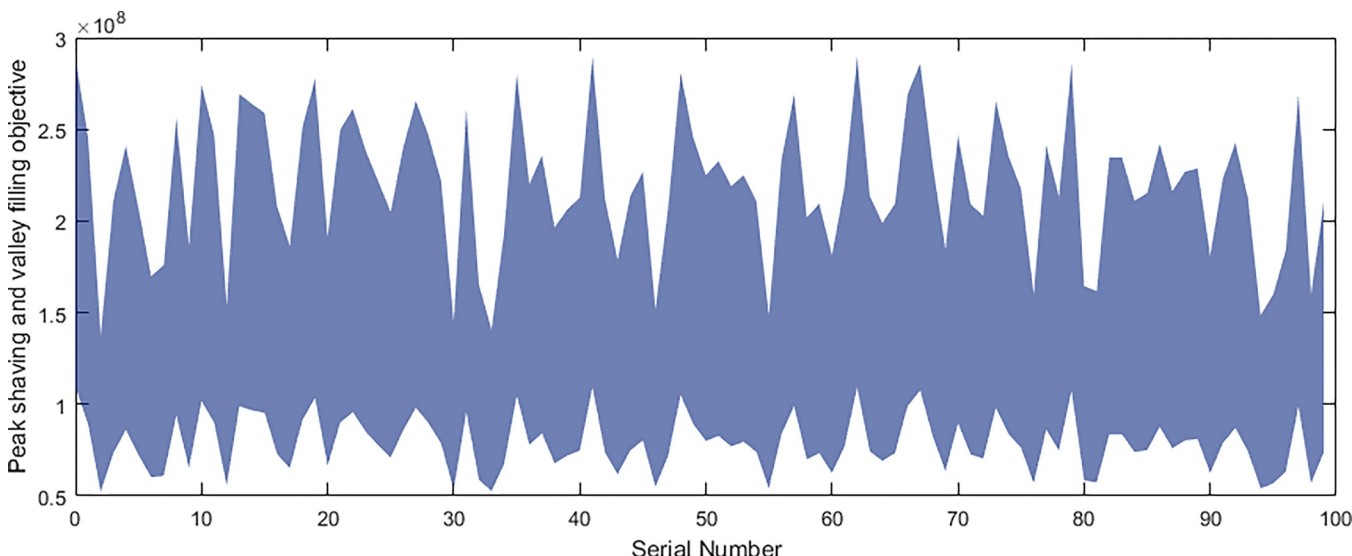

**Fig 18. Peak shaving and valley filling objective.**

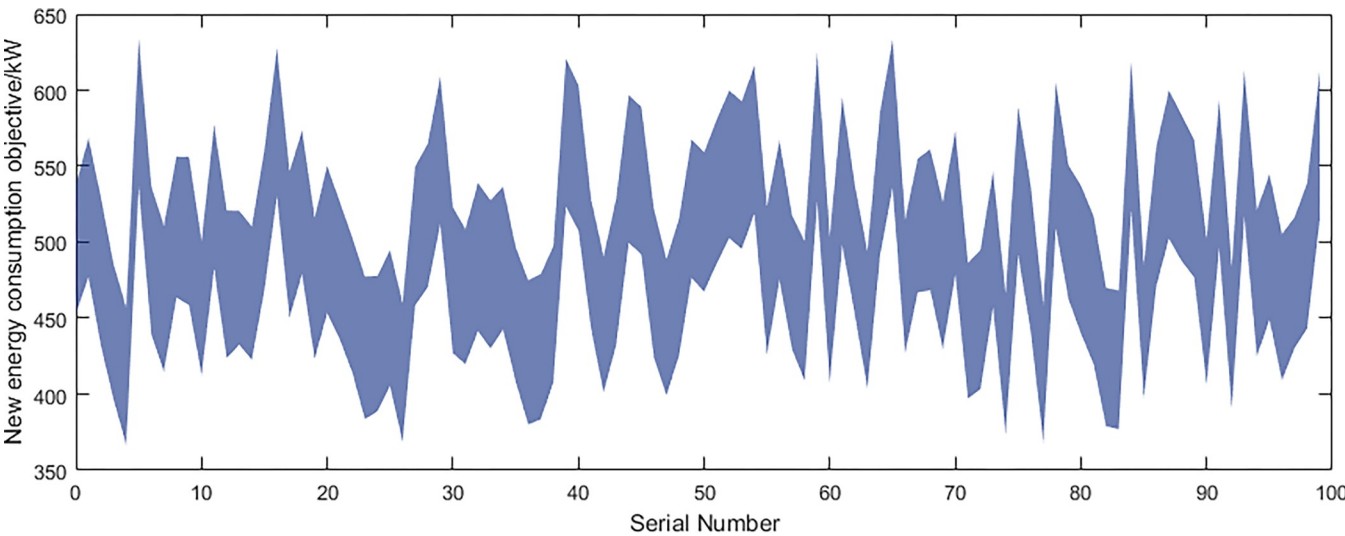

**Fig 19. New energy consumption and integration objective.**

The interval median of the new energy consumption objective function is smaller, but the interval width is larger. This indicates that the introduction of uncertain factors into the interval optimization model enables the most economical solution obtained to pursue better economic performance while also considering the peak shaving and valley filling objectives and the new energy consumption objectives. This further proves that the solution obtained by the interval optimization model can better balance multiple objectives when extreme scenarios are considered, and has stronger robustness.

## Conclusion

The charging behavior of EVs is characterized by flexibility and uncertainty, and a large number of EVs with uncontrolled charging connected to the grid greatly increases the pressure of the grid. Electric vehicles have dual functions of energy storage and energy supply, and can be used as power grid energy storage devices when the vehicle is idle, which provides convenience for power grid peak regulation. In addition, electric vehicles can also be charged and

**Table 3. The objective function values of scheduling schemes considering extreme scenarios.**

| Economic objective/yuan | | Peak shaving and valley filling objective | | New energy consumption objective/kW | |
|---|---|---|---|---|---|
| Interval midpoint | Interval width | Interval midpoint | Interval width | Interval midpoint | Interval width |
| 132,480.42 | 21606.12 | $7.13 \times 10^7$ | $6.39 \times 10^6$ | 449.37 | 429.37 |

**Table 4. The objective function value of the most economical scheme of the deterministic optimization model.**

| Fluctuation coefficient | Economic objective/yuan | | Peak shaving and valley filling objective | | New energy consumption objective/kW | |
|---|---|---|---|---|---|---|
| | Interval midpoint | Interval width | Interval midpoint | Interval width | Interval midpoint | Interval width |
| 0.2 | 102987.52 | 62940.54 | $3.01 \times 10^8$ | $1.94 \times 10^7$ | 565.15 | 462.25 |
| 0.5 | 106403.96 | 76820.54 | $3.08 \times 10^8$ | $4.66 \times 10^7$ | 1479.48 | 1198.54 |
| 0.8 | 116464.34 | 97481.87 | $3.16 \times 10^8$ | $7.08 \times 10^7$ | 2636.96 | 2240.95 |
| 1.0 | 123480.40 | 108165.32 | $3.21 \times 10^8$ | $8.65 \times 10^7$ | 3499.72 | 3024.10 |

**Table 5. The objective function value of the most economical solution for interval optimization model.**

| Fluctuation coefficient | Economic objective/yuan | | Peak shaving and valley filling objective | | New energy consumption objective/kW | |
|---|---|---|---|---|---|---|
| | Interval midpoint | Interval width | Interval midpoint | Interval width | Interval midpoint | Interval width |
| 0.2 | 129513.25 | 13998.58 | $1.58 \times 10^8$ | $1.75 \times 10^7$ | 501.65 | 467.92 |
| 0.5 | 127749.23 | 37999.38 | $1.56 \times 10^8$ | $4.35 \times 10^7$ | 1228.93 | 1458.10 |
| 0.8 | 125933.54 | 65510.11 | $1.69 \times 10^8$ | $7.00 \times 10^7$ | 2243.71 | 2631.80 |
| 1.0 | 124136.86 | 86246.09 | $1.74 \times 10^8$ | $8.02 \times 10^7$ | 3024.10 | 3498.09 |

discharged through the grid to achieve two-way flow of energy, thereby further improving the utilization efficiency of electric energy. As a dispatching resource, EVs can be scheduled for charging and discharging based on time-of-use pricing incentives, and the following conclusions can be drawn:

Firstly, compared to simply exchanging electricity with the external power grid as a smoothing method, using electric vehicles as scheduling resources guides electric vehicle users to charge during low electricity consumption periods and discharge during peak electricity consumption periods, helping the regional power grid suppress load fluctuations and further reducing volatility.

Secondly, EV aggregators can group multiple EVs into an aggregate according to the needs of the grid, so as to achieve the effect of load regulation and peak cutting and valley filling on the power grid. This approach can not only improve the dispatching capacity of the power grid, but also provide economic incentives for EV owners, promoting the popularization of EVs.

Finally, the paper considered the impact of uncertainty factors on the dispatching process, and introduced the concept of interval number to make the model more realistic. The scheduling scheme obtained by interval multi-objective scheduling model is more robust and practical than that obtained by multi-objective scheduling scheme without uncertainty factors.

## Supporting information

**S1 Dataset.**
(XLSX)

**S1 File.**
(7Z)

## Acknowledgments

The author would like to thank State Grid Henan Electric Power Company for providing the data, as well as the laboratory teachers for their assistance in algorithm and writing.

## Author Contributions

**Conceptualization:** Yuanpeng Hua, Weiliang Liu.

**Data curation:** Yuanpeng Hua, Shiqian Wang, Yuanyuan Wang.

**Formal analysis:** Linru Zhang, Weiliang Liu.

**Funding acquisition:** Yuanpeng Hua, Shiqian Wang, Yuanyuan Wang.

**Investigation:** Yuanpeng Hua, Shiqian Wang, Yuanyuan Wang.

**Methodology:** Linru Zhang, Weiliang Liu.

**Project administration:** Yuanpeng Hua, Shiqian Wang, Yuanyuan Wang, Weiliang Liu.

**Software:** Linru Zhang, Weiliang Liu.

**Supervision:** Yuanpeng Hua, Weiliang Liu.

**Validation:** Yuanpeng Hua, Weiliang Liu.

**Visualization:** Linru Zhang.

**Writing – original draft:** Linru Zhang.

**Writing – review & editing:** Yuanpeng Hua, Weiliang Liu.

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
