## [Decision Letter · Decision Letter 0]

21 Aug 2023

PONE-D-23-13822Optimal Dispatching of Regional Power Grid Considering Vehicle Network InteractionPLOS ONE

Dear Dr. Liu,

Thank you for submitting your manuscript to PLOS ONE. After careful consideration, we feel that it has merit but does not fully meet PLOS ONE’s publication criteria as it currently stands. Therefore, we invite you to submit a revised version of the manuscript that addresses the points raised during the review process.

We look forward to receiving your revised manuscript.

Kind regards,

Sheraz Aslam

Academic Editor

PLOS ONE

Journal Requirements:

4. Please ensure that you refer to Figures 2, 3, 4 and 5 in your text as, if accepted, production will need this reference to link the reader to the figure

3. PLOS requires an ORCID iD for the corresponding author in Editorial Manager on papers submitted after December 6th, 2016. Please ensure that you have an ORCID iD and that it is validated in Editorial Manager. To do this, go to ‘Update my Information’ (in the upper left-hand corner of the main menu), and click on the Fetch/Validate link next to the ORCID field. This will take you to the ORCID site and allow you to create a new iD or authenticate a pre-existing iD in Editorial Manager. Please see the following video for instructions on linking an ORCID iD to your Editorial Manager account: https://www.youtube.com/watch?v=_xcclfuvtxQ.

Additional Editor Comments :

Based on reviewers suggestions, I must recommend the article for major revisions.

Reviewers' comments:

Reviewer's Responses to Questions

**Comments to the Author**

1. Is the manuscript technically sound, and do the data support the conclusions?

Reviewer #1: Yes

Reviewer #2: Yes

2. Has the statistical analysis been performed appropriately and rigorously? 

Reviewer #1: Yes

Reviewer #2: Yes

3. Have the authors made all data underlying the findings in their manuscript fully available?

Reviewer #1: No

Reviewer #2: Yes

4. Is the manuscript presented in an intelligible fashion and written in standard English?

Reviewer #1: No

Reviewer #2: Yes

5. Review Comments to the Author

Reviewer #1: This paper developed an EV model in this study that can effectively and reasonably describe the charging and discharging characteristics and behavior of EVs, fully considering the goals and constraints that EVs need to achieve in participating in the power grid dispatch process, introducing uncertainty description, and establishing an interval multi-objective optimization model to make the obtained optimal dispatch scheme more reasonable and practical. I have two comments:

1. There are not enough illustrative pictures to demonstrate the proposed model.

2. The language is poor.

3. The pictures of this paper are coarse and not clear.

Reviewer #2: Comment 1. Can you please update the abstract by incorporating the following information?

a. Details about the EV charging and discharging behavior model established in the paper? How does this model help analyze the dispatching potential?

b. what does the electricity price play as an incentive parameter in the optimization dispatching model? How does it contribute to achieving the objectives of economic efficiency and renewable energy utilization?

Comment 2.

Can you summarize the findings from the research mentioned in the introduction section regarding EV participation in grid dispatch? That can help more the reader to understand the previous work. Furthermore, what are the key takeaways from the previous studies mentioned, and how do they contribute to the understanding of EV-grid interaction?

Comment 3:

When discussing the establishment of the regional power grid scheduling model, provide more details about how EVs are integrated into the model as regular loads that do not participate in scheduling.

Comment 4:

Provide clear and concise explanations of the key insights from Figures 6a and 6b. What do the thermal power, load curve, PV unit power output, and wind turbine unit power output comparisons signify? Emphasize the differences between cases with and without EV participation.

Comments 5:

Elaborate on how the proposed strategy works in detail. Explain the mechanism by which EVs and energy storage devices are charged or discharged during periods of high or low electricity demand, respectively. Describe how this strategy contributes to load curve smoothness and peak-to-valley reduction.

6. PLOS authors have the option to publish the peer review history of their article (what does this mean?). If published, this will include your full peer review and any attached files.

Reviewer #1: No

Reviewer #2: **Yes: **Kaleem Ullah

---

## [Author Response · Author response to Decision Letter 0]

6 Oct 2023

Dear Editors and Reviewers:

Thank you for your letter and for the reviewers' comments concerning our manuscript entitled “Optimal Dispatching of Regional Power Grid Considering Vehicle Network Interaction” (ID: PONE-D-23-13822) . Those comments are all valuable and very helpful for revising and improving our paper, as well as the important guiding significance to our researches. We have studied comments carefully and have made correction which we hope meet with approval. Revised portion are marked in red in the paper (Revise the language issue raised by Reviewer #1 with blue markings), The main corrections in the paper and the responds to the reviewer's comments are as flowing :

Reviewer #1:

Q1:There are not enough illustrative pictures to demonstrate the proposed model.

Response: We are grateful for the suggestion. To be more clearly and in accordance with the reviewer concerns, we have added a figure (Fig 1) to demonstrate the EV model.

Q2:The language is poor.

Response: We apologize for the language problems in the original manuscript. The language presentation was improved with assistance from a native English speaker with appropriate research background.And the modified part has been highlighted in blue.

Q3:The pictures of this paper are coarse and not clear.

Response: Thank you for your comment, and we have re output the figures in the article.

Reviewer #2:

Q1:Can you please update the abstract by incorporating the following information?

a.Details about the EV charging and discharging behavior model established in the paper? How does this model help analyze the dispatching potential?

Response: We are grateful for the suggestion.As suggested by the reviewer, we have added more details of the EV model.The modifications are as follows:

We have revised the original expression : “Initially, an EV charging and discharging behavior model is established, and its dispatching potential is analyzed” to expression : “Initially, based on the user behavior characteristics and charging and discharging characteristics of electric vehicles, a charging and discharging behavior model of electric vehicles was established. Based on the Monte Carlo sampling algorithm, the scheduling upper and lower limits of each scheduling cycle of electric vehicles were described, and the scheduling potential of each scheduling cycle of electric vehicles was obtained”.

b.what does the electricity price play as an incentive parameter in the optimization dispatching model? How does it contribute to achieving the objectives of economic efficiency and renewable energy utilization?

Response: We are grateful for the suggestion.The original wording here is not clear enough,We have revised the original expression : “An optimization dispatching model is then developed, with the electricity price used as an incentive parameter, and the objectives of maximizing the economic efficiency of the regional power grid, optimizing the utilization of renewable energy, and minimizing the steepness of the demand-side power curve.” to expression : “Then, the electricity price is then used as an incentive parameter to guide EV users to charge during periods of low electricity prices and participate in discharge during periods of peak electricity prices. Aiming at the highest economic efficiency, the best consumption effect of new energy and the smoothest demand-side power curve of regional power grid, a three-objective optimal dispatching model was established”.

Q2:Can you summarize the findings from the research mentioned in the introduction section regarding EV participation in grid dispatch? That can help more the reader to understand the previous work. Furthermore, what are the key takeaways from the previous studies mentioned, and how do they contribute to the understanding of EV-grid interaction?

Response: Thank you for your comment, and we have added the following statement in the introduction section : ”The above research results are true and effective, and provided a good reference for the research in this paper. The above paper has conducted a detailed study on the potential of electric vehicle scheduling, fully considering the impact of electric vehicle participation in grid scheduling on the reliability and economy of grid operation, and has achieved good research results. However, the research in the above papers is generally one-sided and does not fully utilize the scheduling potential of electric vehicles, and the final scheduling results do not meet the actual user needs, making practical application difficult ”.

Q3:When discussing the establishment of the regional power grid scheduling model, provide more details about how EVs are integrated into the model as regular loads that do not participate in scheduling.

Response: We are grateful for the suggestion.We have added the following statement in chapter “EVs do not participate in scheduling”:”When electric vehicles do not participate in scheduling, we believe that they start charging when they enter the station, until the charging is completed or the electric vehicle leaves the station”.

Q4:Provide clear and concise explanations of the key insights from Figures 6a and 6b. What do the thermal power, load curve, PV unit power output, and wind turbine unit power output comparisons signify? Emphasize the differences between cases with and without EV participation.

Response: We are grateful for the suggestion. We have revised the original expression “The optimized scheduling results obtained from the participation of EVs in the scheduling can be compared with the simulation results without the participation of EVs. It is evident that EVs have significant advantages in balancing electricity loads and peak shaving. The fluctuation rate of electricity load is reduced by 38.21%. The participation of EVs in the scheduling reduces the output of thermal power generation during peak electricity consumption, while increasing its output during off-peak consumption, ensuring the long-term stable operation of the power plants. The abandonment rate of wind and PV power is reduced by 31.24%.” in chapter “EVs do not participate in scheduling” to expression “By comparing the power generation of thermal power units, it is found that the participation of electric vehicles in scheduling reduces the power generation during peak electricity consumption, while increasing the power during off peak electricity consumption, ensuring the long-term stable operation of thermal power plants. Comparing the load curve, it is found that when electric vehicles participate in scheduling, the fluctuation rate of power load decreases by 38.21% compared to when electric vehicles do not participate in scheduling. Comparing the output power of wind turbines and photovoltaic power generation units, it is found that when electric vehicles participate in scheduling, the abandonment rate of wind and photovoltaic power generation decreases by 31.24% compared to when electric vehicles do not participate in scheduling.he optimized scheduling results obtained from the participation of EVs in the scheduling can be compared with the simulation results without the participation of EVs. It is evident that EVs have significant advantages in balancing electricity loads and peak shaving. The fluctuation rate of electricity load is reduced by 38.21%. The participation of EVs in the scheduling reduces the output of thermal power generation during peak electricity consumption, while increasing its output during off-peak consumption, ensuring the long-term stable operation of the power plants. The abandonment rate of wind and PV power is reduced by 31.24%”.

Q5:Elaborate on how the proposed strategy works in detail. Explain the mechanism by which EVs and energy storage devices are charged or discharged during periods of high or low electricity demand, respectively. Describe how this strategy contributes to load curve smoothness and peak-to-valley reduction.

Response: We are grateful for the suggestion. We have revised the original expression “Firstly, using EVs as dispatching resources to smooth load fluctuations in regional power grids can further reduce volatility compared to simply exchanging power with external grids as a smoothing method” in chapter “Conclusion” to expression “Firstly, compared to simply exchanging electricity with the external power grid as a smoothing method, using electric vehicles as scheduling resources guides electric vehicle users to charge during low electricity consumption periods and discharge during peak electricity consumption periods, helping the regional power grid suppress load fluctuations and further reducing volatility”. 

We would love to thank you for allowing us to resubmit a revised copy of the manuscript and we highly appreciate your time and consideration.

Sincerely.

Weiliang Liu.

---

## [Decision Letter · Decision Letter 1]

15 Jan 2024

Optimal Dispatching of Regional Power Grid Considering Vehicle Network Interaction

PONE-D-23-13822R1

Dear Dr. Liu,

We’re pleased to inform you that your manuscript has been judged scientifically suitable for publication and will be formally accepted for publication once it meets all outstanding technical requirements.

Kind regards,

Sheraz Aslam

Academic Editor

PLOS ONE

Additional Editor Comments (optional):

The authors improved the manuscript based on reviewers and it can be accepted now.

Reviewers' comments:

Reviewer's Responses to Questions

**Comments to the Author**

1. If the authors have adequately addressed your comments raised in a previous round of review and you feel that this manuscript is now acceptable for publication, you may indicate that here to bypass the “Comments to the Author” section, enter your conflict of interest statement in the “Confidential to Editor” section, and submit your "Accept" recommendation.

Reviewer #2: All comments have been addressed

2. Is the manuscript technically sound, and do the data support the conclusions?

Reviewer #2: Yes

3. Has the statistical analysis been performed appropriately and rigorously? 

Reviewer #2: Yes

4. Have the authors made all data underlying the findings in their manuscript fully available?

Reviewer #2: Yes

5. Is the manuscript presented in an intelligible fashion and written in standard English?

Reviewer #2: Yes

6. Review Comments to the Author

Reviewer #2: The authors have adequately addressed all the comments and the paper is now in satisfactory shape.

7. PLOS authors have the option to publish the peer review history of their article (what does this mean?). If published, this will include your full peer review and any attached files.

Reviewer #2: **Yes: **Dr. Kaleem Ullah

---

## [Editor Report · Acceptance letter]

8 Feb 2024

PONE-D-23-13822R1 

PLOS ONE

Dear Dr. Liu, 

I'm pleased to inform you that your manuscript has been deemed suitable for publication in PLOS ONE. Congratulations! Your manuscript is now being handed over to our production team.

Kind regards, 

on behalf of

Dr. Sheraz Aslam 

Academic Editor

PLOS ONE